# Single or pluralistic? The game and balance of China's community governance policy tools

**Hongxun Xiang** [1]*, **Yangfan Bu**[1], **Xunhua Wang**[2]

**1** School of Public Administration, Sichuan University, Chengdu, China, **2** School of Community for the Chinese Nation, Southwest Minzu University, Chengdu, China

* xianghx@stu.scu.edu.cn

## Abstract

### Background

Policy tools embody policy concepts and are essential to achieving policy objectives. The effective allocation of policy tools directly impacts the effectiveness of community governance and determines the modernization process of grassroots governance. We aim to analyze the logic of community governance policy tool selection, and then provide assistance for the modernization of grassroots governance.

### Methods

We selected 100 national and provincial government work reports and 63 policy documents related to community governance during China's "12th Five-Year Plan" to "14th Five-Year Plan" period as analysis samples. And build an analysis framework based on the three dimensions of time, space, and tools. We used Nvivo.20 software for text encoding analysis.

### Results

Based on the model framework, we analyze the results as follows. From the perspective of the time dimension, among the five types of policy tools, the proportion of command-type policy tools used showed a downward trend, from 88.16% in the 12th Five-Year Plan to 83.50% in the 14th Five-Year Plan. However, motivation-type and persuasion-type tools showed an upward trend, rising from 1.34% and 5.26% in the 12th Five-Year Plan period to 3.40% and 8.74% in the 14th Five-Year Plan respectively. The system-change-type policy tools decreased from 1.32% in the 12th Five-Year Plan to 0.97% in the 14th Five-Year Plan. The proportion of capacity-building-type policy tools has gradually increased from 2.63% in the 12th Five-Year Plan to 4.85% in the 14th Five-Year Plan. From the perspective of spatial dimension, apart from command and persuasion policy tools, the usage frequency of the other three types of policy tools in the three major regions all display a "growth-decline-growth" trend. From the perspective of tool dimension, command-type policy tools are dominant in China's community governance, with a cumulative frequency of 1405 times and a high proportion of 81.75%. Apart from command policy tools, persuasive policy tools and

**Data Availability Statement:** All relevant data are within the paper.

**Funding:** The author(s) received no specific funding for this work.

**Competing interests:** The authors have declared that no competing interests exist.

capacity-building policy tools have a relatively high proportion, with usage frequencies of 186 and 78 respectively.

## Conclusions

We found that current community governance policy tools mainly consist of command tools. However, there is a trend towards combining tools such as command, persuasion, incentive, capacity building, and system change in the future. There is a typical contradiction between instrumental rationality and value rationality, indicating an evolution from instrumental rationality to the integration of instrumental and value rationality. This study addresses the conflict of policy tools through rational guidance of values, the rational guarantee of tools, and cooperation to achieve the goal of high-quality development of community governance.

## Introduction

### Background

Community governance aims to address the "pain points" [1], "blockage points" [2], and "difficult points" [3] of the community, meet the growing needs of community residents for a better life [4], and effectively achieve the goal of modernizing the grassroots governance system and governance capacity [5]. Achieving this requires certain ways or means, which cannot be separated from the rational use of policy tools [6]. Policy tools are the methods and measures adopted by public policy actors to achieve public policy goals [7], and they represent a way to achieve a specific policy goal rather than an action plan [8]. In other words, policy tools are an abstract unit of analysis that reflects the coupling of value rationality, instrumental rationality, and institutional rationality [9]. China's community governance policy has transformed policy concepts, from "Good governance and Good governance" to "People-centered" to "New development concepts" and "Satisfying people's pursuit of a better life." [10] Policy objectives have also evolved from "Maintaining good order" to "Equalizing public services" to "Ecological livability" and "Quality of life [11]." As a bridge connecting policy objectives, policy concepts, and policy results, policy tools must combine new development and evolutionary characteristics to reflect the requirements of the times [12].

Currently, international scholars have made significant contributions to the study of policy tools [13], exploring the public choice [14] and policy network approach to policy tool research [15]. However, research on policy tools by Chinese scholars started relatively late and has mainly focused on the concept [16], classification [17], and selection of policy tools [18]. Despite clearer definitions of policy tools, there are still research shortcomings, particularly in the specific application of policy tools, especially in community governance [19].

The policy tool process theory asserts that policy implementation methods and operation plans should vary across different policy stages [20]. It emphasizes the need for comprehensive consideration of value factors, driving forces, and goal-oriented interactions in policy tool analysis [21]. According to this theory, the evolution of policy tools is closely tied to policy objectives and values [22]. At the Sixth Plenary Session of the 16th Central Committee, the Chinese Communist Party introduced the concept of "rural communities," which marked the beginning of China's continuous promotion of community construction. The goals of this effort were "orderly management," "perfect service," and "civilized and peaceful." The concept of "community governance" was first proposed at the 18th National Congress of the

Communist Party of China, which integrated policy concepts into governance at the "micro-cell" level, opening a new chapter in community governance [23]. The Third Plenary Session of the 18th Central Committee of the Communist Party of China put forward the major proposition of "modernization of national governance system and governance capacity," which gradually made "governance modernization" and "efficient governance" the primary policy goals for community governance [24]. At the 19th National Congress of the Communist Party of China, it was pointed out that the country's basic contradiction had transformed into "the contradiction between the people's growing need for a better life and unbalanced and inadequate development." Consequently, the corresponding value concept of community governance evolved into "meeting the people's needs for a better life [25]." Since the implementation of the rural revitalization strategy, promoting the integration and balanced development of urban and rural areas has become the primary goal of development [26]. Therefore, the analysis of community governance policies must consider the time dimension. Based on the above evolutionary stage characteristics, this study divides the policy text into three stages: "2013-2015," "2016-2020," and "2021-2022," to identify the evolution characteristics of rational selection of community governance policy tools over time.

The theory of policy tool design suggests that the selection of policy tools is mainly influenced by the policy environment [27], and focuses on the comprehensive structural analysis of social problems and countermeasures [28]. The broad policy environment includes all factors that affect policy formulation [29], such as the natural environment, social environment, economic environment, human environment, and institutional environment [30]. China's development exhibits distinct regional differences. From a natural environment perspective, the eastern region is primarily composed of urban communities due to its proximity to the coast and predominantly flat terrain, whereas the central and western regions have complex geology and frequent natural disasters, resulting in mainly scattered rural communities [31]. In terms of the economic environment, the eastern region's contribution to the national GDP is 54.22%, the central region accounts for 28.32%, and the western region accounts for 17.10%. Consequently, the uneven economic development across regions leads to differing material bases for community development [32]. From a social and cultural standpoint, the per capita education level and resident quality are higher in the eastern region, which contributes significantly to community construction [33]. In terms of the institutional environment, the eastern region is a pioneer in institutional reform and innovation, with smart governance being a successful pilot measure in community governance [34]. Based on these differences in the economy, society, humanities, and institutions, this study analyzes the differences in community governance policy tools from a spatial perspective, exploring the commonalities and unique features of the eastern, central, and western regions.

Thus, this study comprehensively integrates process theory and design theory of policy tools, aiming to construct a three-dimensional analysis framework based on time, space, and tools. This framework will be utilized to identify the rational choice of community policy tools' evolution trend. Additionally, the study aims to model and interpret community governance policy texts from a policy tools perspective, seeking to explore the internal logic of community governance policy tools' evolution and expand the scope of China's research on policy tools. Ultimately, this research will aid in modernizing China's grassroots governance system and improving governance capacity.

## Literature review

Policy Tools, also known as Tools of Government [35] or Governing Instruments [36]. In academia, the conceptual expression of policy tools is not uniform enough. Hughes OE. argues

that a policy tool is an expression of government behavior [37]; Salamon LM. believes that policy tools are methods or approaches to solving public problems through collective action [38]; Chinese scholar Zhang CF. believes that policy tools are means or mechanisms to translate policy goals into concrete actions and make policy goals [39]; Mao SL. believes that policy tools are the means by which the government can realize its management functions [40]. At present, the research on policy tools mainly focuses on the classification of policy tools. As early as the 60s of the 20th century, German economist Kirschen ES. and others explored the classification of policy tools from the field of economics [41]. At the same time, American political scientists Dahl RA, and Lindblom CE. also made similar explorations in the field of political science, using a two-way classification method to briefly divide policy tools into regulatory tools and non-regulatory tools [42]. Since the 80s, the study of policy tool classification has reached a climax. Hood C. divided policy tools into four categories: informational tools, authoritative tools, financial tools, and formal organizational tools [43]; Bemelmans-Videc, Marie-Louise, et al. divides policy tools into three categories: legal tools, economic tools, and communication tools [44]; Howlett M, Ramesh M, Perl A. divide policy tools into three categories: voluntary tools, mandatory tools and mixed tools [45]; McDonnell LM, Elmore RF. divide policy tools into five categories: imperative tools, remuneration tools, function extension tools, authority restructuring tools, and exhortation tools [46]; Schneider AL, Ingram HM, editors. divide policy tools into five categories: authoritative tools, incentive tools, capacity-building tools, symbolic tools, and learning tools [47]. On the basis of absorbing the existing classifications, Chinese scholars have subdivided policy tools into authoritative, inducing [48], competent, learning, persuasive [49], commanding, capacity building, systematic transformation, voluntary, mixed, and compulsory [50]. It also further explores the selection of policy tools, including the influencing factors of policy tool selection and the model construction of policy tool selection. For example, scholars such as Ding Hua believe that the selection of policy tools needs to comprehensively consider the attributes of tools and the external environment [50].

It is apparent that while there is a relatively rich body of research on policy tools, most of the existing studies focus on the tools themselves, with limited research on their practical application, particularly in the context of community governance. Chinese scholar Yei L. H. suggests that choosing appropriate policy tools to resolve the challenge of fragmented property rights of urban public resources is an effective way to address the issue of public resources [51]. Xu J. Y. argues that the use of policy tools is directly related to the effectiveness of community governance [52]. Drawing on data from the community governance pilot area from 2011 to 2018, Kong N. N. identifies the development trend of community network governance in its emerging stage [53]. Shan F. F. finds that the use of community governance policy tools is highly unbalanced, as demonstrated in the analysis of the "Opinions on Strengthening and Improving Urban and Rural Community Governance" [54]. However, most existing studies focus on the macro-level definition, components, selection, and evaluation methods of community governance policy tools, with content limited to specific perspectives such as public resources, digitization, and community security. As such, there is a lack of in-depth analysis of the internal logic and evolution mechanism of community governance policy tools in China. Therefore, it is essential to conduct a systematic analysis and explanation of the evolution trend of China's community governance policy tools and their internal contradictions.

## Materials and methods

### Sample selection

Judging from the results of data published on the websites of relevant departments at all levels, the concept of community governance first appeared in policy documents after the Third

Plenary Session of the 18th Central Committee [55]. Although there were previously relevant contents of community management, they were scattered in various documents and did not form a system. Due to the lack of policy texts related to community governance, and the documents issued by different levels of departments have different characteristics [4]. Therefore, the following principles were adopted when selecting sample files. First, the sample period is from 2013 to 2022; Second, the sample mainly comes from policy documents at or above the provincial level, including five-year planning documents, special planning documents, government work reports, etc.; Third, because some provinces have not issued special policies for community governance, the provincial sample range is three provinces in each of the three major regions of the eastern, central and western regions delimited by the state, and the government work report of the sample province is the main line. The papers number these texts according to the principle of "region-period-order" to form a library of policy texts required for research.

The policy texts analyzed in this study were collected from the official websites of local governments and civil affairs departments. The government work reports from 2013 to 2022 and relevant policy documents from the 12th to 14th Five-Year Plan periods were selected to illustrate the temporal evolution of community policy tools in China (China began implementing community governance in 2013). To present the spatial variation of community governance policy tools, this study sampled 3 provincial-level administrative units from each of the eastern, central, and western regions of China, for a total of 9 sample areas. The sample extraction did not consider the difference in the number of administrative units in the region (The study refers to the division standards of the Fourth Session of the Sixth National People's Congress in 1986 and the Fifth Session of the Eighth National People's Congress in 1997 to divide China into eastern, western, and central regions. Sample extraction takes into account the influencing factors such as the regional economy, society, and culture). To ensure representativeness, the study selected national government work reports, civil affairs development plans, representative provincial government work reports, and other policy documents related to community governance. As a result, the representativeness of policy documents is fully ensured from the perspectives of time, space, and content. A total of 163 policy documents are selected as research objects, and they are categorized and numbered according to regional differences, forming a database of community governance policy texts (Table 1).

## Measures

As this study primarily examines policy texts, a content analysis approach was employed. This involved using NVIVO.20 text coding software to conduct coding analysis, which enabled the conversion of non-quantitative policy text into quantitative data for ease of understanding. Furthermore, utilizing the dimensions of time, space, and tools, the paper analyzes various

**Table 1. Sample library of community governance policy texts.**

| Type | National | | Eastern Region | | Central Region | | Western Region | | Total |
| --- | --- | --- | --- | --- | --- | --- | --- | --- | --- |
| | | | Fujian, Guangdong, Hebei | | Anhui, Hubei, Heilongjiang | | Gansu, Sichuan, Yunnan | | |
| | Number | Quantity | Number | Quantity | Number | Quantity | Number | Quantity | |
| Government Work Report | N-A | 10 | East-A | 30 | Center-A | 30 | West-A | 30 | 100 |
| National economic development planning, Civil affairs development planning, Community governance development planning | N-B | 18 | East-B | 15 | Center-B | 15 | West-B | 15 | 63 |
| Total | | 45 | | 45 | | 45 | | 45 | 163 |

types of community governance policy tools, along with their differences over time and in different locations. By doing so, this study delves deeper into the temporal and spatial trends and characteristics of community governance policies, specifically within the policy tools dimension. Additionally, the paper explores the internal selection logic and public value orientation of policy tools in the same time and space (Fig 1).

**Time dimensions.** The concept and objectives of community governance policies exhibit phased characteristics over time. Similarly, government policy tools are influenced by these changing policy concepts and objectives [49], resulting in phased characteristics. Thus, this study examines policy documents on community governance from 2013 to 2022, as community governance in China was initiated in 2013. These policy documents include national economic development planning, government annual work reports, civil affairs development plans, opinions on strengthening and improving the governance of urban and rural communities, and opinions on strengthening the grassroots governance system and modernizing governance capacity. Through analyzing the logic and trends in the selection of policy tools for community governance in China over time, this study aims to explore the rational allocation of policy tools for future community governance.

**Spatial dimensions.** In terms of spatial dimensions, the two positive principles of China's central and local relations emphasize the need to uphold the unified leadership of the central authorities while also giving full play to local initiative and enthusiasm. The community governance policy profoundly reflects this relationship between the central and local governments in China. In 2013, the state introduced the concept of community governance, and subsequent provincial governments started incorporating community governance in their new planning and work reports. In 2017, the central government issued opinions on strengthening and improving the governance of urban and rural communities, and provinces such as Sichuan, Guangzhou, and Hunan successively issued implementation opinions on the same. The

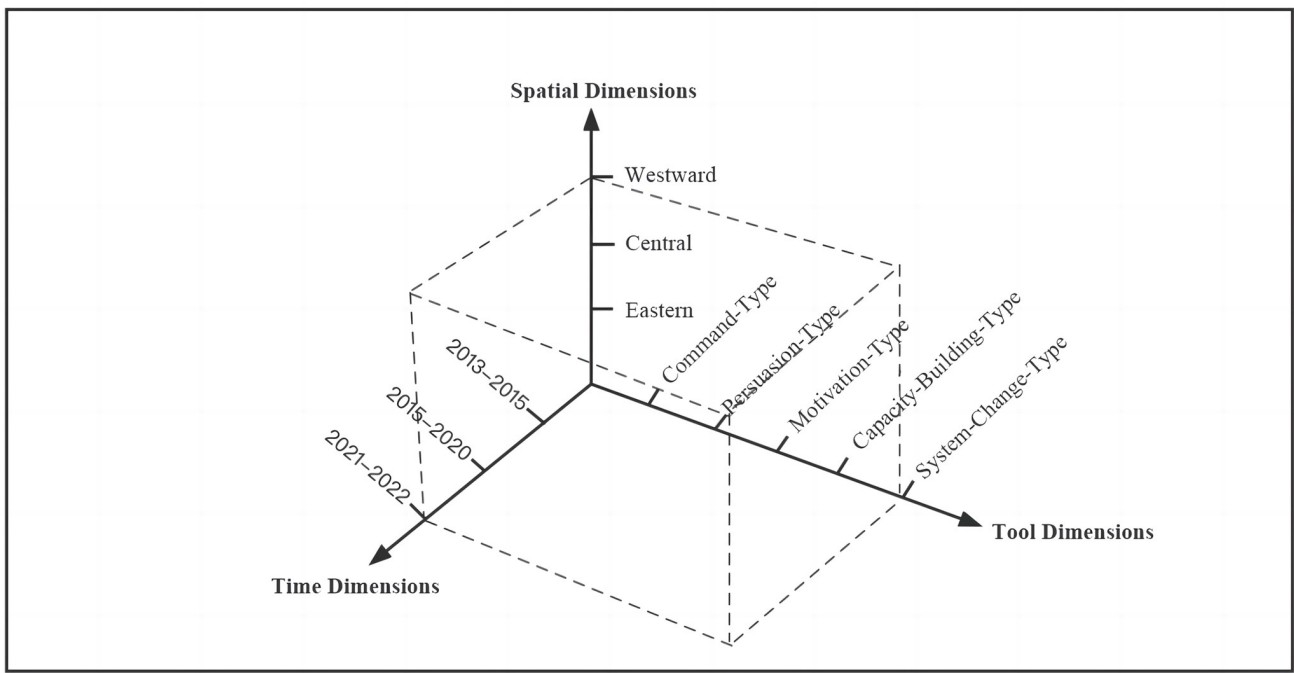

**Fig 1. Analysis framework.**

relationship between the central and local governments affects the identity of local policies, while competition among local governments leads to regional differences [26]. For instance, in the 14th Five-Year Plan for Urban and Rural Community Governance (Service), Guangdong Province emphasizes improving the urban and rural community governance system, continuously enhancing the level of urban and rural community governance, and strengthening the governance infrastructure of urban and rural communities. Hunan Province, on the other hand, calls for the provision of basic public services to communities, while Sichuan Province's planning theme is centered around "development" and "governance," with special plans for community development. As a result, local governments inevitably make differentiated decisions based on the economic, social, and human environment, under the unified leadership of the central government. Thus, understanding the trends in these regional differences and how they promote the modernization of grassroots governance systems and capabilities requires exploration from the spatial dimension.

**Tool dimensions.** In terms of tool dimensions, policy tools are the means to achieve an end, and the completion of the goal of community governance modernization requires the appropriate combination of policy tools. There is much debate in academic circles about policy tools, and this study adopts the classification criteria proposed by scholars McDonnell, L. M., Elmore, R. F. [46] and Schneider, A., and Ingram, H. [56] based on the research needs of community governance. The community governance policy tools are divided into five types: command-type, persuasion-type, motivation-type, capacity-building-type, and system-change-type. Command-type policy tools are based on coercive and legitimate authority, requiring unconditional obedience from governance objects. They are mainly in the form of rules, standards, orders, requirements, and systems, which are manifested as "must", "requirements" and "implementation" in policy texts. They have the advantages of high efficiency and low cost. Persuasion-type policy tools encourage policy targets to take specific actions by giving great significance to specific things. They are mainly achieved through guiding demonstration, assessment and evaluation, public opinion propaganda, and showing them in policy texts such as "promotion", "encouragement", and "typical". Although they lack coercion, they have strong value guidance. Motivation-type policy tools induce policy targets to make relevant behaviors through material or spiritual rewards. They are mainly achieved through evaluation, commendation, job promotion, etc., and are reflected in "rewards" and "promotions". These tools are the ones that are most in line with the assumption of economic man. Capacity-building-type tools ensure the realization of policy goals by cultivating capacity and improving quality. They are mainly achieved through platform construction, education, learning, etc., which are manifested as "training" and "improvement" in policy texts. They have long-term characteristics. Systematic-change-type policy tools emphasize changing the existing organizational structure and institutional arrangements to adapt to new policy goals. They are manifested in organizational structure adjustment, industrial structure change, spatial layout optimization, etc., and the policy text is displayed through "reshaping" and "soundness". They are the embodiment of organizational change theory in policy tools.

## Results

### Text encode

In this paper, a total of 163 samples of policy texts such as national and provincial five-year plans, special plans, and government work reports were selected as the text library. Therefore, the study uses Nvivo.20 text encoding software to encode the files of the policy text library, so that the qualitative policy text can be converted into quantitative node data. In the coding process, according to the classification of policy tools in the analysis model, the coding structure

of "tree node-subnode-reference point" and "reference point-subnode-tree node" is adopted. When encountering policy texts that are difficult to classify, the research adopts the semantic analysis method, combined with the context content of the text to make comprehensive judgments. Specifically, according to the theory related to the theme, the coding nodes are identified, and according to the actual situation of the theory and policy text, the policy tools are divided into five categories: command-type, motivation-type, persuasion-type, system-change-type, and capacity-building-type. Secondly, subnodes are established under the tree node, and then the text reflecting each dimension is set as the reference point by line-by-line coding, and then its hierarchical classification is programmed into the child node and the tree node, and finally the coding structure of the tree node-child node-reference point is formed. Finally, when the text content can be compiled into multiple child nodes, the method of semantic judgment is used to determine its true meaning in combination with the context and then compiled into the corresponding node.

## Reliability and validity

To ensure the credibility and validity of the final coding results, two researchers were tasked with randomly selecting uncoded policy texts from Eastern, Central, and Western regions for encoding. The encoded results were imported into NVIVO 20, and the consistency of the coding was verified through the software's "code comparison query" function. Furthermore, to ensure the reliability of the results, each policy text was coded by two researchers, with weight given to the number of codes that agreed or disagreed with each other. If the percentage was greater than 70%, it was considered to have high confidence(A consistency coefficient of 0.0-0.20 indicates extremely low consistency, a consistency coefficient of 0.21-0.40 indicates general consistency, a consistency coefficient of 0.41-0.60 indicates medium consistency, a consistency coefficient of 0.61-0.80 indicates a high degree of consistency, and a consistency coefficient of 0.81-1 indicates almost complete consistency.). The results of the theoretical fullness test indicated that the coding conformance percentage and the kappa coefficient of coding coverage remained above 0.6, which was considered highly consistent. The agreement percentage was 93%. Therefore, the coding in this paper is valid and has a high degree of confidence.

## Descriptive statistics

In this study, policy text was encoded to form a coding structure of "tree node-child node-reference point." Analysis of the coding results revealed that the reference point was proportional to the child node, and the frequency was proportional to the ratio under the premise of keeping the tree node unchanged. That is, the higher the frequency and ratio, the higher the preference for public selection (Table 2) [23]. Firstly, overall, the proportion of command-type policy tools decreased gradually over time, with the frequency of use dropping from 88.16% in the 12th Five-Year Plan to 85.05% in the 13th Five-Year Plan and then dropping to 83.50% in the 14th Five-Year Plan. This reflects the progress made in the modernization of the national governance system and governance capacity, with management thinking gradually shifting to governance thinking [57]. Although persuasion-type, motivation-type, capacity-building-type, and system-change-type tools fluctuated, they showed an overall upward trend, with their proportions gradually increasing from 2.63%, 5.26%, 1.32% and 2.63% to 3.74%, 7.48%, 1.87% and 1.87%, respectively. This reflects a trend towards diversification of governance policy tools, in line with the emphasis on "multi-subject participation in community governance" in recent years [58]. Secondly, from a regional perspective, the eastern region showed obvious regional characteristics, with the proportion of command-type and motivation-type policy tools increasing, while the frequency of use of other policy tools decreased year by year. This is

**Table 2. Results of policy text encoding for community governance.**

| File type, Time, Frequency and Policy instrument type. | | Command-type | Motivation-type | Persuasion-type | System-change-type | Capacity-building-type |
|---|---|---|---|---|---|---|
| Central and provincial data aggregation | The 12th Five-Year Plan (2011-2015) | 67/88.16% 420/80.31% | 2/2.63% 7/1.34% | 4/5.26% 60/11.47% | 1/1.32% 6/1.15% | 2/2.63% 30/5.74% |
| | The 13th Five-Year Plan (2016-2020) | 91/85.05% 623/82.85% | 4/3.74% 10/1.33% | 8/7.48% 78/10.37% | 2/1.87% 8/1.06% | 2/1.87% 33/4.39% |
| | The 14th Five-Year Plan (2021-2025) | 86/83.50% 362/82.09% | 2/1.94% 15/3.40% | 9/8.74% 48/10.88% | 1/0.97% 1/0.23% | 5/4.85% 15/3.40% |
| Eastern Region | The 12th Five-Year Plan (2011-2015) | 128/74.85% | 3/1.75% | 27/15.79% | 2/1.17% | 11/6.43% |
| | The 13th Five-Year Plan (2016-2020) | 194/81.51% | 3/1.26% | 31/13.03% | 2/0.84% | 8/3.36% |
| | The 14th Five-Year Plan (2021-2025) | 155/82.01% | 8/4.23% | 23/12.17% | 0/0.00% | 3/1.59% |
| Central Region | The 12th Five-Year Plan (2011-2015) | 143/81.71% | 1/0.57% | 16/9.14% | 2/1.14% | 13/7.43% |
| | The 13th Five-Year Plan (2016-2020) | 215/84.31% | 2/0.78% | 19/7.45% | 4/1.57% | 15/5.88% |
| | The 14th Five-Year Plan (2021-2025) | 104/81.25% | 4/3.13% | 13/10.16% | 0/0.00% | 7/5.47% |
| Western Region | The 12th Five-Year Plan (2011-2015) | 149/84.18% | 3/1.69% | 17/9.60% | 2/1.13% | 6/3.39% |
| | The 13th Five-Year Plan (2016-2020) | 214/82.63% | 5/1.93% | 28/10.81% | 2/0.77% | 10/3.86% |
| | The 14th Five-Year Plan (2021-2025) | 103/83.06% | 3/2.42% | 12/9.68% | 1/0.81% | 5/4.03% |

in line with the goal of "efficient governance" in the eastern region, using the strong pulling power of command-type policy tools and the strong impetus of motivation-type policy tools to ensure the scheduled promotion of grassroots governance modernization [59]. The frequency of use of command-type policy tools in the western region has decreased, while the frequency of persuasion-type, motivation-type, capacity-building-type, and system-change-type policy tools has increased. This is due to the implementation of targeted poverty alleviation and rural revitalization strategies in the western region, with the tilt of national policies leading to a diversified trend in policy tools [60]. The frequency of use of various policy tools in the central region fluctuated up and down, reflecting frequent changes in policy objectives in this region, with various instruments showing a "rise-fall-rise" cycle change pattern [61].

## Analysis framework

**Analysis of time dimensions.** Among the five types of policy tools, the proportion of command-type policy tools used showed a downward trend, from 88.16% in the 12th Five-

Year Plan to 83.50% in the 14th Five-Year Plan. However, motivation-type and persuasion-type tools showed an upward trend, rising from 1.34% and 5.26% in the 12th Five-Year Plan period to 3.40% and 8.74% in the 14th Five-Year Plan respectively (Fig 2). This shows that China's governance level is constantly improving, paying more attention to democracy, civil rights, and participation in consultation, emphasizing more community residents' autonomy and villagers' autonomy, changing the traditional command-led policy action logic, and then using more diversified policy tools to guide and encourage policy targets to achieve policy goals. The system-change-type policy tools decreased from 1.32% in the 12th Five-Year Plan to 0.97% in the 14th Five-Year Plan. System-change-type policy tools represent the organizational change theory in policy tools, emphasizing changing the existing organizational structure and institutional arrangements to adapt to new policy goals, mainly manifested as organizational structure adjustments. Since the reform and opening-up, China has carried out eight large-scale party and government institutional reforms, with the last reform in 2018, forming an institutional structure with comprehensive party leadership, clear responsibilities and powers, responding to social concerns, and reasonable power allocation. China's organizational structure has entered a stage where organizational stability is the main goal, and as the structure becomes more perfect, the frequency of using system-change-type policy tools will gradually decrease. The proportion of capacity-building-type policy tools has gradually increased from 2.63% in the 12th Five-Year Plan to 4.85% in the 14th Five-Year Plan, which reflects the importance attached to talents in grassroots governance, especially after the opinions on strengthening the modernization of the grassroots governance system and governance capacity were proposed. Introducing, cultivating, and retaining community talents has become a key task in community governance. Provinces such as Sichuan, Zhejiang, and Jilin have included dedicated chapters in the "14th Five-Year Plan" for urban and rural community development and governance planning, proposing to "promote the construction of community talent teams".

**Analysis of spatial dimensions.** Analysis from a holistic perspective shows that, apart from command and persuasion policy tools, the usage frequency of the other three types of policy tools in the three major regions all display a "growth-decline-growth" trend (Fig 3).

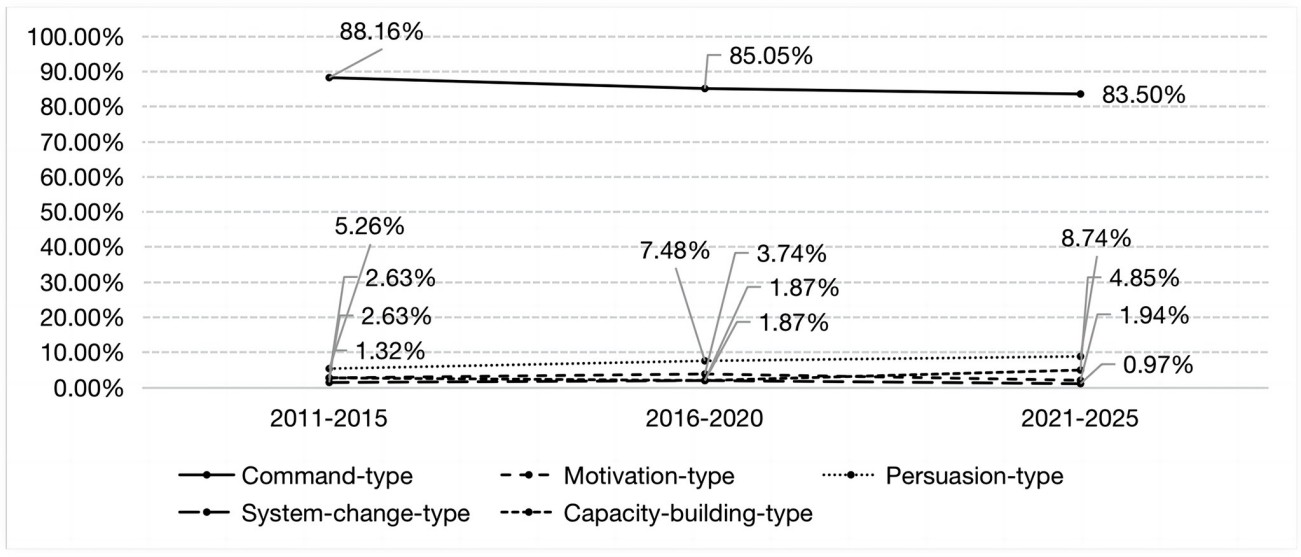

**Fig 2. Time dimension frequency chart.**

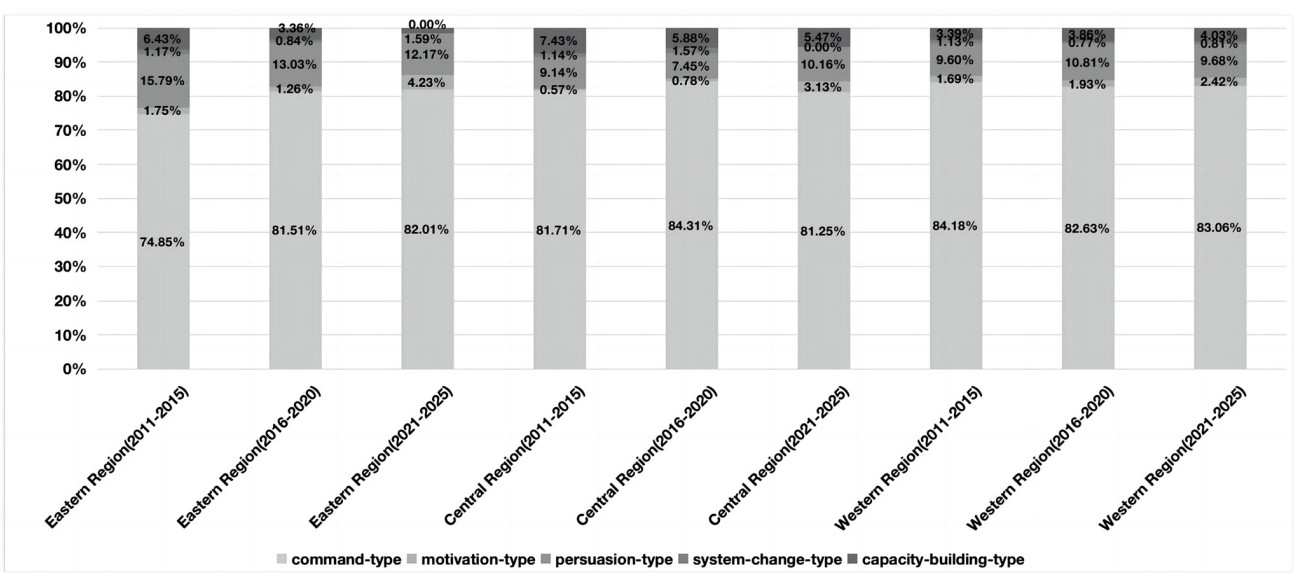

**Fig 3. Spatial dimension frequency chart.**

Overall, command policy tools have the largest proportion, with a reference point ratio of over 80%. The other four policy tools have a proportion of less than 10%, with only the persuasion tool accounting for 11.47%, 10.37%, and 10.88% respectively. This indicates the urgency of high-quality, efficient, effective, and modern community governance in the modernization strategy of national governance, which has prompted provincial and lower-level governments to gradually abandon incentive and capacity-building tools that have been widely used and effective in the early stages, and turn to command policy tools. In terms of reference points, there are significant spatial differences in the policy tools used in community governance in the three major regions, but the reference points for command policy tools are the highest among all policy tools, with a minimum gap of 5 times and a maximum gap of 10 times. Therefore, it can be seen that command policy tools are the most important in community governance policy and are the most commonly used by provincial governments.

Looking at each region internally, apart from command policy tools, the policy tools used in community governance in the eastern region are mainly capacity-building and persuasion tools. The use of persuasion tools has remained stable, ranging from 13.03% to the current 12.17%. However, capacity-building and system change tools have shown a downward trend, with a decrease from 6.43% to 1.59% and from 1.17% to 0.84%, respectively. In addition, the government work reports during the 14th Five-Year Plan period did not reflect any content related to system change tools. Incentive tools have shown a "growth-slight decrease-rapid growth" characteristic, rising from 1.75% to 4.23%. This indicates that the wording of the government's texts in the eastern region tends to be moderate, promoting grassroots governance workers to quickly promote community governance performance through "logical reasoning" and other methods, while conducting extensive training for grassroots governance personnel, which has greatly improved their community governance and innovation capabilities. From an intra-regional perspective, except for command policy tools, the policy tools used for community governance in the central region show significant temporal differences. In terms of increase, during the 12th Five-Year Plan period, command tools (81.71%) and persuasion

tools (15.79%) were mainly emphasized, during the 13th Five-Year Plan period, system reform tools (0.43%) and incentive tools (0.19%) were emphasized, and during the 14th Five-Year Plan period, incentive tools (2.35%) and persuasion tools (2.69%) were mainly emphasized. It can be seen that the use of policy tools in central provinces strives for innovation during each five-year plan period, and there are differences between them. This may have two consequences: firstly, effective policy tools that have been proven to be ineffective in practice in promoting the effectiveness of community governance are forced to be abandoned; secondly, to understand the central government's intentions and learn from advanced experiences, different policy tools are adopted at different stages. The fact proves that the lack of continuity in policy tools can also lead to poor community governance effectiveness, resulting in the phenomenon of inadequate modernization of community governance in the central region [24]. In the Western region, apart from a decrease in the use of command and system reform policy tools, the other three types of policy tools have gradually increased. This indicates that the community governance policies in the western region have timely adjusted the proportion of policy tool usage according to the actual changes at the grassroots level, to obtain sustained innovation power and improve governance effectiveness.

**Analysis of tool dimensions.** Through analysis of the coding files, it is not difficult to see that command-type policy tools are dominant in China's community governance, with a cumulative frequency of 1405 times and a high proportion of 81.75%. This indicates that since the proposal of community governance, the government has mainly relied on command-type policy tools to promote the development of community governance, which is consistent with the urgency of modernizing grassroots governance. Specifically, the overall frequency of command-type policy tool usage is relatively stable, maintaining around 80%. In terms of different regions, the proportion of command-type policy tools in the eastern region has gradually increased from 74.85% to 82.01%; in the central region, the frequency of command-type policy tools shows a "decrease-increase-decrease" cyclical trend, increasing from 81.71% to 84.31% and then decreasing to 81.25%; in the western region, the proportion of command-type policy tools is gradually decreasing, from 84.18% to 83.06% (Fig 4). This is consistent with the actual situation in each region. The eastern region, at the forefront of China's economic and institutional reforms, is forced to choose command-type policy tools to ensure the realization of the two-step strategy of grassroots governance modernization. The central region, due to its lower level of modernization, has a large fluctuation in specific governance objectives, which leads to the instability of its policy tools. The western region, as the main support target of national macro policies such as western development, targeted poverty alleviation, and rural revitalization, has made significant progress in improving grassroots governance level and governance capacity with the advancement and completion of various strategies. The degree of acceptance and participation of community residents in policies is increasing, and a diversified pattern of community governance participation has formed, leading to a gradual decrease in the frequency of mandatory policy measures such as command-type policies. Apart from command policy tools, persuasive policy tools and capacity-building policy tools have a relatively high proportion, with usage frequencies of 186 and 78 respectively. Each region is consistent with the overall trend, with persuasive policy tools being used 81, 48, and 87 times in the east, central, and west, respectively, while capacity-building policy tools are used 22, 35, and 21 times. This reflects the changing policy concept in Chinese communities, from "good order and governance" to "people-centered" to "new development concept" and "meeting the pursuit of a better life for the people," continuously highlighting the importance of the mass line. Correspondingly, the frequency of mandatory policy tools with high coerciveness has gradually decreased, while the frequency of persuasive and incentive policy tools with higher guidance has gradually increased. In addition, talent team building has always been an important

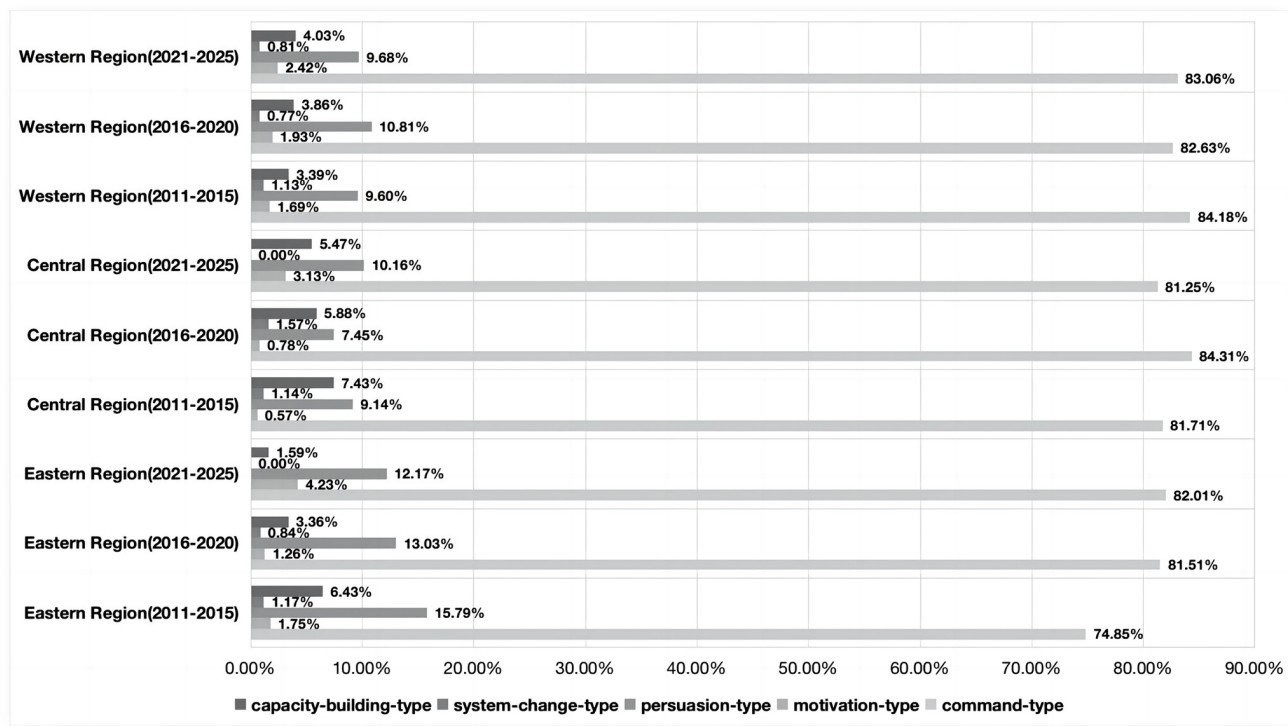

**Fig 4. Tool dimension frequency chart.**

measure for China to solve grassroots governance issues. In 2021, the General Office of the Communist Party of China Central Committee and the General Office of the State Council issued the "Opinions on Accelerating the Revitalization of Rural Talents," corresponding to the increasing use of capacity-building policy tools in various regions.

## Discussion

The 6th Plenum of the 19th CPC Central Committee proposed a three-step strategy for socialist modernization, which requires significant progress in modernization by the 100th anniversary of the Party's founding, the basic realization of modernization of governance and governance capabilities by 2035, and comprehensive realization of national governance modernization by the 100th anniversary of the founding of the People's Republic of China. Against the backdrop of the three-step strategy for national governance modernization, a two-step development strategy for grassroots governance has been proposed, namely, the basic realization of modernization of grassroots governance by 2025 and the comprehensive realization of modernization of grassroots governance by 2035. Therefore, the modernization of grassroots governance must achieve its goals 15 years ahead of the modernization of national governance. This has led to the use of policy tools with higher degrees of compulsion to ensure the timely realization of the modernization strategy of grassroots governance, rather than following the conventional policy path with democracy, participation, and consultation as its core. This has also resulted in a conflict between the value rationality centered on service and governance, and the tool rationality centered on compulsion and command.

## The rational choice of policy tools reflects the current situation of policy tool singularity based on tool rationality as the benchmark

Tool rationality emphasizes means rather than value pursuit, and has a strong goal orientation, which means that it is oriented towards policy goals and uses appropriate policy tools to ensure the achievement of policy goals, without paying attention to the policy ideas behind the goals [62]. Currently, the main goals of China's urban and rural community policies include efficient governance, modern governance, and rural revitalization. Achieving these community policy goals requires corresponding policy tools to ensure their success. Efficient governance and modern governance at the community level are important components of modernizing the grassroots governance system, following the modernization of the national governance system. Rural revitalization is a major strategic deployment after the completion of the comprehensive poverty alleviation campaign, and is a necessary choice for the contradiction between China's dual development pattern and the goal of integrated and balanced development. Based on the three-step strategy of modernizing national governance and the two-step strategy of modernizing grassroots governance, modernizing grassroots governance needs to be completed 15 years ahead of modernizing national governance, and modernizing community governance, as an important part of grassroots governance, needs to be completed ahead of the time point for modernizing grassroots governance. In addition, rural revitalization also proposes a three-step development strategy, corresponding to the modernization of national and grassroots governance. The three-step strategy requires achieving the goal of agricultural and rural modernization by 2035. Compared with urban community governance, rural community governance faces prominent problems such as low quality of organizational leaders, weak party-building leadership, serious youth outflow and lack of governance subjects, mismatched rights and responsibilities in towns and villages, and weak economic foundations. The urgency of time and the complexity of community governance require the use of command-type policy tools with greater mandatory effect to ensure that the goal of modernizing urban and rural community governance by 2035 is achieved as scheduled, thereby guaranteeing the promotion of the modernization of grassroots governance and national governance strategies. Therefore, under tool rationality, community governance policy tools present a current state of using single policy tools based on command-type policy tools as a benchmark.

## The rational choice of policy tools presents a trend of policy tool diversification guided by value rationality

Value rationality focuses on the values that behaviors themselves can represent rather than the results of the chosen behaviors, emphasizing the values represented by policy tools themselves, that is, using certain substantial and specific value concepts to judge the rationality of policy tool usage [63]. In community governance, value rationality is mainly reflected in policy concepts, which guide the use of policy tools and contain the value pursuits of policy concepts. Based on the community governance text corpus, this article extracts the changes in community policy concepts in various government work reports, national economic development plans, development plans for civil affairs units, and related policy documents from 2013 to 2022. From the perspective of historical changes, during the 12th Five-Year Plan period, community governance policy concepts focused on the people-oriented dimension, with few other policy concepts, and relatively single policy concepts. Since the 13th Five-Year Plan, multiple concepts have emerged and coexist, expanding from a people-oriented approach to integrating governance and new development concepts that include precise, efficient, collaborative, and high-quality development. The development trend of governance policy concepts from singularity to diversity in value levels necessarily requires the evolution of policy tools at the

practical level toward diversification. The core of the new development concept is innovation, coordination, greenness, openness, and sharing. Innovation requires attention to the role of technology, talents, and other factors in community governance, and the use of government policy tools that focus on capacity building and incentives. Coordination requires clarification of responsibility mechanisms, introducing multiple subjects to participate in community governance, and using persuasion and systemic change government policy tools comprehensively. Green requires protection of the ecological environment, and should combine mandatory, incentive, persuasion, and capacity-building policy tools based on the construction of the soft and hard environment in the community to create a green and environmentally friendly living environment. Openness and sharing require everyone's participation and benefit, and require strengthening the construction of the community talent team through capacity-building, systemic change, and mandatory governance policy tools to gather all forces to build a harmonious and safe community and to benefit residents from development, achieving community-wide sharing. The core of integrating policy concepts is integration thinking, which means promoting the integration of governance logic in community governance through diversified governance methods and resources [64]. Diversified governance methods require the diversified development of policy tools, focusing not only on the use of mandatory and systemic change governance tools but also on the introduction of persuasive, incentive, and capacity-building tools. The people-oriented concept requires putting people at the center, emphasizing democracy and consultation, and gradually reducing the frequency of using mandatory policy tools in community governance while increasing the use of incentive, persuasive, and capacity-building tools. Based on this, the policy concept has shifted from a single "people-oriented" approach to a diverse "new development concept" and "integrated policy concept." The policy tools are also showing a trend towards diversification, with the frequency of traditional policy tools such as mandatory and systemic change gradually decreasing, while the frequency and proportion of incentive, persuasive, and capacity-building policy tools are constantly increasing. Community policy tools are developing toward diversity.

## The contradiction and reconciliation of tool rationality and value rationality: The rational game and balance of policy tools

Value rationality and instrumental rationality have both identity and struggle. The value rationality orientation with policy ideas as the core and the instrumental rationality orientation with policy goals as the core of community governance present the dual characteristics of conflict and harmony. The harmony of community policy tools is mainly reflected in three aspects: firstly, the value rationality of policy tools is the spiritual driving force of instrumental rationality, and the value rationality reflected by community policy ideas is the guide for policy tool selection, showing the trend of diverse policy tool development. Secondly, instrumental rationality is the realistic support of value rationality. The policy tool pattern dominated by the command type is currently the choice that meets the actual situation of the community and is the tool guarantee for realizing value rationality. Thirdly, instrumental rationality and value rationality are the same in urban and rural community governance practices. Although the trend of diversification of value rationality conflicts with the current situation of instrumental rationality singularity, the singularity of tool use is a process and stage of diversified values and a necessary approach. The conflict of community policy tools is highlighted in two aspects. Firstly, the contradiction between the pursuit of diversified values and the singularity of tool use. As policy ideas expand towards new development concepts, precision, efficiency, and collaboration, the tools that guarantee the implementation of ideas should present a trend of diversification along with the expansion of ideas, but the singularity of tool use contradicts this

trend. Secondly, the conflict between policy ideas centered on democracy and consultation and policy tools dominated by command type. With the transformation from management to governance, values such as democracy, participation, consultation, and sharing have become the value choices of current policies, which conflict with the mandatory nature of community policy tools. China has experienced the transformation from management to governance, and community governance is currently in this period of transition. Although the current stage shows a trend of using command-type policy tools, there is also a trend of diversified development. With the deepening of community governance, a diversified policy tool pattern will inevitably form.

## Conclusion

The Marxist theory of contradictions holds that contradictions have both identity and struggle, which coexist in the development of things. At present, community governance presents a contradiction between the diversification trend of value-oriented rationality and the singularity of tool-oriented rationality. However, contradictions are a stage of development, and unity is the inevitable trend after contradictions. The rational choice of community governance policy tools will move towards diversification. The implementation of community governance policy goals depends on the specific implementation of policy tools, and the rational allocation of policy tools depends on the valuable guidance of policy concepts. The Central Committee of the Communist Party of China and the State Council jointly issued an opinion on strengthening and improving urban and rural community governance. Provincial-level administrative units in various regions have also successively issued the 14th Five-Year Plan for Urban and Rural Community Governance (Service), which integrates the new development concept, policy concept, and people-oriented concept, and implements policy goals such as efficient governance, modern governance, and rural revitalization. Comprehensive use of policy tools such as command, incentive, guidance, capacity building, and system change ensures the continuous development of community governance and the achievement of the goal of modernizing grassroots governance as planned.

This study focuses on the evolutionary trends and characteristics of the current rational selection of community policy tools, and although the current tool usage status is dominated by command-type tools, there is a contradiction between value-oriented rationality and tool-oriented rational choice, but there is also unity. The rational choice of policy tools shows a trend of diversified development, which is also the conclusion of this study based on the analysis of 163 policy texts from the national and nine provincial levels. Under the guidance of diverse policy concepts and governance policy goals, community governance policy tools show both singularity and diversification. This evolutionary process of policy tools can not only ensure the timely completion of community policy goals but also respond to people's urgent need for a better life. In the future, community governance should further play the guiding role of value-oriented rationality, the guaranteeing role of tool-oriented rationality, promote the integration of tool-oriented rationality and value-oriented rationality, and cooperate to support the modernization of urban and rural community governance by using a combination of punches.

Overall, this study used text coding software to code and analyze 163 policy texts to identify the contradictions and trends in current community governance. Compared with previous studies, this study has some innovations. The first is to shift the analysis of policy texts from qualitative to quantitative analysis to make them more persuasive. The second is to expand the field of analysis of policy tools, which in the past focused more on the tools themselves, while research introduced tools into community governance. Finally, the construction of the three-

dimensional model is not only analyzed from the policy tool itself, but also combined with the time dimension and spatial dimension to make the research more explanatory. However, the study also has several limitations. Firstly, the number of policy texts analyzed is relatively low, and selecting all policy texts might have been more scientifically sound. Secondly, although coding was done using coding tools, the coding process is still influenced by human factors and cannot guarantee objectivity completely. Nevertheless, this study comprehensively used software tools to analyze policy texts and scientifically validated the results, complying with standard research methods and having scientific conclusions that can effectively analyze existing problems. In the future, if the range of texts and coding frequency can be expanded, the study will be even more reasonable.

## Acknowledgments

The data used in this study come from the policy documents publicly released by the government, and the text coding adopts Nvivo 20. We are very grateful for the sharing of policy documents on the government websites of various regions, and thanks for the technical support provided by Nvivo software. In addition, we are also very grateful to Xinyi Huang, Jing Wang, Ziwei Lin, Can Yang, etc. for their help with text encoding.

## Author Contributions

**Conceptualization:** Hongxun Xiang.

**Data curation:** Hongxun Xiang, Yangfan Bu, Xunhua Wang.

**Formal analysis:** Hongxun Xiang.

**Methodology:** Hongxun Xiang, Xunhua Wang.

**Software:** Hongxun Xiang, Yangfan Bu, Xunhua Wang.

**Supervision:** Hongxun Xiang, Xunhua Wang.

**Visualization:** Yangfan Bu.

**Writing – original draft:** Hongxun Xiang.

**Writing – review & editing:** Yangfan Bu, Xunhua Wang.

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
