## [Decision Letter · Decision Letter 0]

30 May 2023

PONE-D-23-12250Single or Pluralistic? The Game and Balance of China's Community Governance Policy Tools：A study of policy documents.PLOS ONE

  Dear Dr. xiang,

Thank you for submitting your manuscript to PLOS ONE. After careful consideration, we feel that it has merit but does not fully meet PLOS ONE’s publication criteria as it currently stands. Therefore, we invite you to submit a revised version of the manuscript that addresses the points raised during the review process based on the reviewers' comments.

We look forward to receiving your revised manuscript.

Kind regards,

Jinlong Liu

Academic Editor

PLOS ONE

Reviewers' comments:

Reviewer's Responses to Questions

**Comments to the Author**

1. Is the manuscript technically sound, and do the data support the conclusions?

Reviewer #1: Yes

Reviewer #2: Yes

Reviewer #3: Partly

2. Has the statistical analysis been performed appropriately and rigorously? 

Reviewer #1: Yes

Reviewer #2: Yes

Reviewer #3: Yes

3. Have the authors made all data underlying the findings in their manuscript fully available?

Reviewer #1: No

Reviewer #2: Yes

Reviewer #3: Yes

4. Is the manuscript presented in an intelligible fashion and written in standard English?

Reviewer #1: Yes

Reviewer #2: Yes

Reviewer #3: Yes

5. Review Comments to the Author

Reviewer #1: Thank you for giving me the opportunity to read "Single or pluralistic? The game and balance of China's community governance policy tools". This paper analyzes the tool selection of community governance policy based on China’s work reports and policy documents during the 12th to 14th five-year period of China. This topic is worthwhile. I think this topic provides an important contribution to the literature. The analytical framework of the three dimensions of time, space, and tools is of great significance to the modernization of grassroots governance in China. The manuscript has some potential. However, the manuscript needs further refinement before it can be accepted for publication. The reviewers list some specific comments that may help the authors to further improve the quality of the manuscript. Please consider the specific comments listed below:

1. Suggest that the author add some illustrations or tables to clarify your point of view and make your article more attractive.

2. The manuscript still has some occasional grammatical errors, especially the articles "the", "a" and "an" are missing in many places, besides these minor problems, please check the spelling. Also, some sentences are too long to be read easily. It is recommended to change to a shorter sentence that is easier to read.

3. Results and discussion section. The two sections are also well structured and organized. However, it is best to discuss further how your findings differ from past work.

Reviewer #2: The topic of this paper is innovative and has practical value. It is a good paper with sufficient demonstration.There are still grammar and spelling errors in the English version of this article. Please revise it carefully. Reference materials, citing literature from scholars outside of China, need to be added, especially those from authoritative authors who are systematic and groundbreaking in policy tools. Figure 1 is not clear enough and needs to be remade.

Reviewer #3: 1. The literature review is incomplete, especially the literature review for the classification of policy tools policy tools. The article only includes competent, learning, persuasive, command, induce,

capacity building, systemic transformation, voluntary, mixed, and coercive. There are other categories that have not been mentioned and it needs to be supplemented.

2. The study selected 163 policy texts including national government work reports, civil affairs development plans, representative provincial government work reports and other policy documents related to community governance. However, the process of screening 163 policies has not been presented. The policy collection process needs to be demonstrated.

3. The text encoding is too simple, please supplement the process of text encoding completely

4. To ensure the reliability of the results, two researchers jointly encode a policy text, the percentage was greater than 70%, it was considered to have high confidence. Is the reliability of this method too low? Maybe need to supplement the explanation of the reliability of this method.

5. The discussion is not closely related to the quantitative analysis of the previous 163 policy texts. The discussion between “The Homogenization of Policy Tools Based on the Criterion of Instrumental Rationality” and “The Trend Towards Diversification of Policy Tools Guided by Value Rationality ”cannot reflect the conclusions drawn from the policy texts well. The analysis content of the policy text is not closely related to the discussion. It is recommended to combine the discussion in the future with the analysis content of the previous policy text.

6. PLOS authors have the option to publish the peer review history of their article (what does this mean?). If published, this will include your full peer review and any attached files.

Reviewer #1: No

Reviewer #2: **Yes: **liu bangfan

Reviewer #3: No

---

## [Author Response · Author response to Decision Letter 0]

8 Jun 2023

Response to Reviewers

Dear Editor Liu, 

We appreciate you and the reviewers for your precious time in reviewing our paper and providing valuable comments. It was your valuable and insightful comments that led to possible improvements in the current version. The authors have carefully considered the comments and tried our best to address every one of them. We hope the manuscript after careful revisions meet your high standards. The authors welcome further constructive comments if any.

Below we provide the point-by-point response. All modifications in the manuscript have been highlighted in blue.

Sincerely, 

Hongxun Xiang 

xianghx@stu.scu.edu.cn

Ph.D., School of Public Administration

Sichuan University

Response to Reviewer 1

[Comment 1] Have the authors made all data underlying the findings in their manuscript fully available?

Response: Thank you for your valuable comments to us. We reorganized the study data and graphed them to be added to the text. For example, Figure 2 (Pg10, Ln349), Figure 3 (Pg11, Ln399), Figure 4 (Pg12, Ln440). Make the data in the manuscript more adequate.

[Comment 2] Suggest that the author add some illustrations or tables to clarify your point of view and make your article more attractive.

Response: Thank you for your valuable comments to us. We have added tables or illustrations from the temporal dimension (Pg10, Ln349), the spatial dimension (Pg11, Ln399), and the tool dimension (Pg12, Ln440) to make the data display more intuitive and the article more attractive.

[Comment 3] The manuscript still has some occasional grammatical errors, especially the articles "the", "a" and "an" are missing in many places, besides these minor problems, please check the spelling. Also, some sentences are too long to be read easily. It is recommended to change to a shorter sentence that is easier to read.

Response: Thank you for your valuable comments to us. We re-examined the grammar of the entire text, especially the articles "the", "a", and "an", and corrected a total of 32 grammatical errors. In addition, we have rewritten overly long sentences to make them simpler and easier to read. Specific changes can be found in a marked copy of the manuscript.

[Comment 4] Results and discussion section. The two sections are also well structured and organized. However, it is best to discuss further how your findings differ from past work.

Response: Thank you for your valuable comments to us. We have revised the discussion section to better correspond to the previous policy text encoding. In addition, we have added some notes to the conclusion section to explore how the current findings differ from previous ones. The first is to shift the analysis of policy texts from qualitative to quantitative analysis to make them more persuasive. The second is to expand the field of analysis of policy tools, which in the past focused more on the tools themselves, while research introduced tools into community governance. Finally, the construction of the three-dimensional model is not only analyzed from the policy tool itself but also combined with the time dimension and spatial dimension to make the research more explanatory (Pg10, Ln605-621).

Response to Reviewer 2

[Comment 1] There are still grammar and spelling errors in the English version of this article. Please revise it carefully.

Response: Thank you for your valuable comments to us. We re-examined the grammar of the entire text, especially the articles "the", "a", and "an", and corrected a total of 32 grammatical errors. In addition, we have rewritten overly long sentences to make them simpler and easier to read. Specific changes can be found in a marked copy of the manuscript.

[Comment 2] Reference materials, citing literature from scholars outside of China, need to be added, especially those from authoritative authors who are systematic and groundbreaking in policy tools.

Response: Thank you for your valuable comments to us. We rewrote the literature review and added references. The focus is on completing the previous lack of foreign scholars, especially pioneering scholars, including Hughes OE., Salamon LM., Kirschen ES., Dahl RA., Hood C., Howlett M., Schneider AL., etc. Specific changes can be found in a marked copy of the manuscript (Pg3, Ln84-114).

[Comment 3] Figure 1 is not clear enough and needs to be remade.

Response: Thank you for your valuable comments to us. We redrew the picture to ensure its clarity of the picture (Pg5, Ln182). 

Response to Reviewer 3

[Comment 1] The literature review is incomplete, especially the literature review for the classification of policy tools policy tools. The article only includes competent, learning, persuasive, command, induce, capacity building, systemic transformation, voluntary, mixed, and coercive. There are other categories that have not been mentioned and it needs to be supplemented.

Response: Thank you for your valuable comments to us. We rewrote the literature review and added references. The focus is on completing the previous lack of foreign scholars, especially pioneering scholars, including Hughes OE., Salamon LM., Kirschen ES., Dahl RA., Hood C., Howlett M., Schneider AL., etc. Specific changes can be found in a marked copy of the manuscript (Pg3, Ln84-114).

[Comment 2] The study selected 163 policy texts including national government work reports, civil affairs development plans, representative provincial government work reports and other policy documents related to community governance. However, the process of screening 163 policies has not been presented. The policy collection process needs to be demonstrated.

Response: Thank you for your valuable comments to us. We've added a new selection process for policy texts, which are as follows. Judging from the results of data published on the websites of relevant departments at all levels, the concept of community governance first appeared in policy documents after the Third Plenary Session of the 18th Central Committee. Although there were previously relevant contents of community management, they were scattered in various documents and did not form a system. Due to the lack of policy texts related to community governance, the documents issued by different levels of departments have different characteristics. Therefore, the following principles were adopted when selecting sample files. First, the sample period is from 2013 to 2022; Second, the sample mainly comes from policy documents at or above the provincial level, including five-year planning documents, special planning documents, government work reports, etc.; Third, because some provinces have not issued special policies for community governance, the provincial sample range is three provinces in each of the three major regions of the eastern, central and western regions delimited by the state, and the government work report of the sample province is the main line. The papers number these texts according to the principle of ”region-period-order” to form a library of policy texts required for research (Pg4, Ln134-151).

[Comment 3] The text encoding is too simple, please supplement the process of text encoding completely.

Response: Thank you for your valuable comments to us. We've added a new description of the coding process. In this paper, a total of 163 samples of policy texts such as national and provincial five-year plans, special plans, and government work reports were selected as the text library. Therefore, the study uses Nvivo.20 text encoding software to encode the files of the policy text library, so that the qualitative policy text can be converted into quantitative node data. In the coding process, according to the classification of policy tools in the analysis model, the coding structure of ”tree node-subnode-reference point” and ”reference point-subnode-tree node” is adopted. When encountering policy texts that are difficult to classify, the research adopts the semantic analysis method, combined with the context content of the text to make comprehensive judgments. Specifically, according to the theory related to the theme, the coding nodes are identified, and according to the actual situation of the theory and policy text, the policy tools are divided into five categories: command-type, motivation-type, persuasion-type, system-change-type, and capacity-building-type. Secondly, subnodes are established under the tree node, and then the text reflecting each dimension is set as the reference point by line-by-line coding, and then its hierarchical classification is programmed into the child node and the tree node, and finally the coding structure of the tree node-child node-reference point is formed. Finally, when the text content can be compiled into multiple child nodes, the method of semantic judgment is used to determine its true meaning in combination with the context and then compiled into the corresponding node (Pg7, Ln251-270).

[Comment 4] To ensure the reliability of the results, two researchers jointly encode a policy text, the percentage was greater than 70%, it was considered to have high confidence. Is the reliability of this method too low? Maybe need to supplement the explanation of the reliability of this method.

Response: Thank you for your valuable comments to us. We've added a description of the consistency ratio. According to the practice of humanities and social sciences, a consistency coefficient of 0.0~0.20 indicates extremely low consistency, a consistency coefficient of 0.21~0.40 indicates general consistency, a consistency coefficient of 0.41~0.60 indicates medium consistency, a consistency coefficient of 0.61~0.80 indicates a high degree of consistency, and a consistency coefficient of 0.81~1 indicates almost complete consistency. The formulation in the text has been revised to make it more reliable (Pg8, Ln279-283).

[Comment 5] The discussion is not closely related to the quantitative analysis of the previous 163 policy texts. The discussion between “The Homogenization of Policy Tools Based on the Criterion of Instrumental Rationality” and “The Trend Towards Diversification of Policy Tools Guided by Value Rationality ”cannot reflect the conclusions drawn from the policy texts well. The analysis content of the policy text is not closely related to the discussion. It is recommended to combine the discussion in the future with the analysis content of the previous policy text.

Response: Thank you for your valuable comments to us. We have restructured the discussion section to better align with the policy text encoding data described earlier. The description is as follows. First, the current status of simplification of policy tools is based on the current results of text encoding. From the coding results, the proportion of command-type policy tools far exceeds the sum of other tools, and the current policy tools are mainly imperative. Second, the trend of diversification of policy instruments is derived through the analysis of the time dimension. Although the current command-type policy tools are the mainstay, the proportion of other policy tools has increased, so it shows a diversified development trend. Specific changes can be found in a marked copy of the manuscript. In addition, we have added some notes to the conclusion section to explore how the current findings differ from previous ones. The first is to shift the analysis of policy texts from qualitative to quantitative analysis to make them more persuasive. The second is to expand the field of analysis of policy tools, which in the past focused more on the tools themselves, while research introduced tools into community governance. Finally, the construction of the three-dimensional model is not only analyzed from the policy tool itself but also combined with the time dimension and spatial dimension to make the research more explanatory (Pg10, Ln605-621).

---

## [Editor Report · Decision Letter 1]

2 Jul 2023

Single or Pluralistic? The Game and Balance of China's Community Governance Policy Tools：A study of policy documents.

PONE-D-23-12250R1

Dear Dr. Xiang,

We’re pleased to inform you that your manuscript has been judged scientifically suitable for publication and will be formally accepted for publication once it meets all outstanding technical requirements.

Kind regards,

Jinlong Liu

Academic Editor

PLOS ONE
---

## [Editor Report · Acceptance letter]

23 Aug 2023

PONE-D-23-12250R1 

Single or pluralistic? The game and balance of China’s community governance policy tools 

Dear Dr. xiang:

I'm pleased to inform you that your manuscript has been deemed suitable for publication in PLOS ONE. Congratulations! Your manuscript is now with our production department. 

Kind regards, 

on behalf of

Professor Jinlong Liu 

Academic Editor

PLOS ONE